

# Reconstructing lateral migration rates in meandering systems; a novel Bayesian approach combining OSL dating and historical maps

Cindy Quik & Jakob Wallinga

Soil Geography and Landscape Group, Wageningen University, The Netherlands

*Correspondence to:* Cindy Quik. Address: P.O. Box 47, 6700AA, Wageningen. Email: cindy.quik@wur.nl.

**Abstract.** Identifying lateral migration rates of meandering rivers is relevant both for fluvial geomorphology and
to support river management. Lateral migration rates for contemporary meandering systems are often reconstructed
based on sequential remote sensing images or historical maps, however the time frame for which these sources are
available is limited and hence likely to represent fluvial systems subjected to human influence. Here, we propose
to use scroll bar sequences as an archive to look further back in time using optically stimulated luminescence
(OSL) dating of sand-sized quartz grains. We develop a modelling procedure for the joint Bayesian analysis of
(OSL) dating results and historical map data. The procedure is applied to two meanders from the Overijsselse
Vecht, a medium-sized sand-bed river in The Netherlands. We obtained 9 samples for OSL dating from scroll bars,
and combined OSL dating results with historical map data for the period 1720 – 1901 AD. The procedure we
propose here incorporates strengths of both data types for reconstructing fluvial morphodynamics over longer
timeframes. Using an iterative modelling approach, we translate spatial uncertainty of historical maps into
temporal uncertainty of channel position required for Bayesian deposition modelling. Our results indicate that
meander formation in the Overijsselse Vecht system started around 1400 AD, and lateral migration rates were on
average 2.6 and 0.9 m/y for the two investigated bends, until river channelization around 1900 AD.

Keywords: optically stimulated luminescence, Bayesian chronological modelling, OxCal, poor bleaching, fluvial
geomorphology, Holocene.

## 1 Introduction

Rivers are one of the world's most important geomorphic agents and represent highly dynamic earth surface
systems (Vandenberghe and Maddy, 2000; Güneralp and Rhoads, 2011). Meandering rivers represent the most
common river type and can be found all over the world (Hooke, 2013). The mobility of their course has a significant
effect on sediment transport and deposition, floodplain development and landscape change (Kleinhans and Van
den Berg, 2011). Related consequences for river management, hazards and biodiversity pose a practical need to
understand the timing and speed of lateral migration (Hooke, 2013; Grabowski et al., 2014; Lespez et al., 2015).

To determine whether changes in meander position and morphology have taken place it is often needed to
consider relatively long time scales (Hooke, 2013). To establish recent lateral migration rates and reconstruct the
planform of contemporary fluvial systems, sequential aerial photographs or topographical maps are often used
(e.g. Pišút, 2002; Uribelarrea et al., 2003; Hooke and Yorke, 2010; Eekhout et al., 2013). Unfortunately, the lack
of historical sources of this type limits long-term reconstructions of channel position, and reconstructions based



on older maps may be grossly affected by spatial uncertainties and mapping inaccuracies. Moreover, as historical

maps are generally only available for cultivated areas, the fluvial systems displayed are likely to be affected by man. Such influence can be either through direct interference with the river system (e.g. Hesselink et al., 2003; Frings et al., 2009), or through land-use which may change the hydrograph (e.g. Candel et al., submitted) as well as sediment input (e.g. De Moor et al., 2008; Hoffmann et al., 2009). Consequences may be large, as shown in a highly-influential paper by Walter and Merritts (2008), who argued that present-day meandering form of gravel-

bedded mid-Atlantic streams in the USA is not a natural phenomenon, but rather an artefact caused by mill ponds. Hence, to understand dynamics of fluvial systems and anthropogenic impacts on those systems, there is a need to extend the period of reconstruction to earlier periods, which requires reconstruction methods that do not rely solely on historical sources and/or remote sensing.

To resolve depositional ages and geomorphological process rates, the use of optically stimulated luminescence

(OSL) dating to obtain absolute chronologies of fluvial deposits is becoming widespread (Wallinga, 2002a; Lian and Roberts, 2006; Rittenour, 2008). In addition, OSL is increasingly used to date young (<1000 years) sediments as part of Late-Holocene studies (Madsen and Murray, 2009). Yet, application of OSL dating to reconstruct lateral migration rates has hardly been attempted, with the notable exception of Rodnight et al. (2005) who used OSL dating to reconstruct lateral migration rates of the Klip river in South Africa during the past 1000 years. Such

limited application of OSL dating for this purpose may well be related to challenges faced when applying luminescence dating to young fluvial sediments. Succesfull age determination requires complete resetting of the OSL signal in at least part of the grains; a prerequisite that may not be met due to limited light exposure during subaquaous transport of grains in a turbid river.

Bayesian analysis of depositional sequences allows the construction of more precise and robust chronologies

by combining ages obtained through a dating method with prior informaton on the order of events (Bronk Ramsey, 2008). Following pioneering work by Rhodes et al. (2003), Bayesian analysis is now increasingly applied to sequences of OSL ages, often in combination with ages obtained through other methods (e.g. [14]C: Dreibrodt et al., 2010; U-series: Clark-Balzan et al., 2012; Dendrochronology: Wallinga et al., 2012), or even multiple methods (e.g. Shanahan et al., 2013; Hobo et al., 2014; Guérin et al., 2017).

Several studies have combined OSL dating with evidence derived from historical maps and documentation as independent age control (e.g. Ballarini et al., 2003; Medialdea et al., 2014), age constraints (e.g. Nielsen et al., 2006; Madsen et al., 2007; Tamura et al., 2011; Kunz et al., 2014), or for historical contextualisation of dating results (e.g. Clemmensen et al., 2007). However, direct integration of OSL dates with legacy data provided by historical evidence into a single geochronology, i.e. where historical evidence is used to help determine the

geochronology, is underexplored. An exception to this is the study by Hobo et al. (2014), who reconstructed floodplain sedimentation of the Dutch river Waal using Bayesian age-depth models based on heavy metal age estimates, OSL dates and age constraints derived from historical maps.

Here we combine OSL data and historical map evidence using Bayesian chronological modelling to reconstruct planform development and lateral migration rates of the Overijsselse Vecht, a medium-sized sand-bed river in The

Netherlands. Our aim is to develop a modelling procedure using the Bayesian depositional sequences tool of OxCal (Bronk Ramsey, 2008) for the joint analysis of OSL dating results and historical map data, that (1) allows quantitative analysis of historical maps in geochronological research and (2) supports the development of robust fluvial chronologies with quantified uncertainty that expand into larger timeframes. With these methods we




determine the age of point bar deposits along the Overijsselse Vecht and the migration rate of two meanders that developed during the past 500 years. This information provides the geochronological basis for a palaeohydrological study on the Overijsselse Vecht (Candel et al., submitted).

## 2 Study area

The Overijsselse Vecht is a medium-sized river originating in Nordrhein Westfalen in Germany and enters the
Netherlands from the East near the town of Coevorden. This rain-fed river has a catchment size of 3785 km$^2$ and a rather uniform valley gradient on Dutch territory of $1.4 * 10^{-4}$ (Wolfert and Maas, 2007). According to measurements in the period 1995-2015 of the discharge station in Mariënberg (just upstream from our study area) the average discharge and mean annual flood discharge amount to 22.8 and 160 m$^3$ s$^{-1}$ respectively. Average annual rainfall in the region amounts to 800-875 mm and average monthly temperatures vary between 4.5-5.0°C in
January and 22.5-23.0°C in July (KNMI, 2017).

The Overijsselse Vecht was channelized in the late nineteenth and early twentieth century. The original river length of 90 km on Dutch territory was reduced to 60 km by the cut off of 69 meanders (Wolfert and Maas, 2007). Weirs were installed to prevent vertical erosion of the river bed, and groynes and revetments were constructed to prevent lateral migration of the river channel (Wolfert et al., 1996). In recent decades, water managers have
initiated several projects to restore the physical and ecological functioning of the river, which posed the need for increased understanding of river morphodynamics (e.g. Wolfert et al., 1996, 2009; Maas et al., 2007; Viveen et al., 2009) and ecological potential (e.g. Duursema, 2004).

The floodplain of the Overijsselse Vecht is flanked on both sides by extensive areas with coversand ridges and drift-sand dunes, overlying fluvioperiglacial deposits of Pleistocene age (Ter Wee, 1966, 1979; Kuijer and Rosing,
1994; geomorphological map of the Netherlands [1:50.000], for details refer to Koomen and Maas, 2004). Its floodplain is very narrow (Fig. 1a) and many meanders appear to be laterally confined by stable valley edges. For this study we focused on two meander bends with large amplitudes that reach outside the valley edge, named Junner Koeland and Prathoek (Fig. 1).





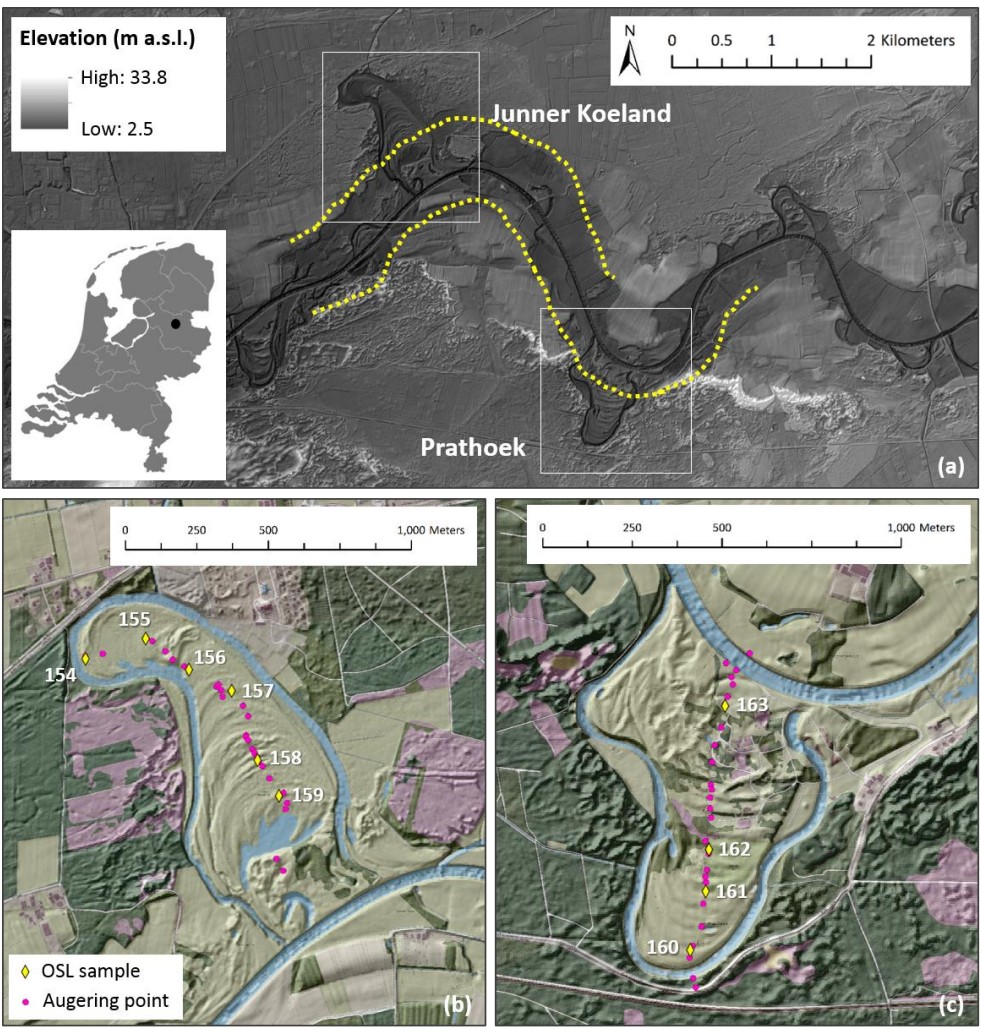

**Figure 1.** Location of the study area (a), showing the two meander bends Junner Koeland (b) and Prathoek (c). Dotted line in (a) indicates the valley edge. M a.s.l. = metres above sea level. White numbers along the coring transects in (b) and (c) indicate OSL sample locations by abbreviated sample code (all should be preceded by NCL-2415). Samples NCL-2415154–162 were collected in point bar deposits, sample NCL-2415163 was collected in a terrace remnant. Lithological cross sections and lithogenetic interpretation are provided in Candel et al. (submitted). Digital elevation model (AHN2, horizontal resolution 5 m, vertical resolution 0.2 m): AHN (2018); Van Heerd and Van't Zand, (1999). Topography (3200 pixels/km): OpenTopo, Van Aalst (2017).

## 3 Methods

### 3.1 Lithological survey

For each meander bend a lithogenetic cross section was constructed based on hand-corings from meander base to apex (Fig. 1bc), with a core obtained at every high and low point in the point-bar and swale topography. In a separate paper, Candel et al. (submitted) provide details on these cross sections (including detailed facies





descriptions and an explanation of the lithogenetic interpretation), and use the information to reconstruct palaeodischarges. Here we concentrate on those aspects that are directly relevant to the reconstruction of channel position throughout the lifetime of the meander. Most of the cores reached a depth of 4 to 6 metres, penetrating the full point-bar deposits (extremely fine to coarse sand) including the channel lag (medium fine to coarse sand

with gravel), which occurred at depths varying from 2.5 to 4 m below the surface. Some aeolian reworking occurred locally, as evident from the DEM (Fig. 1). The lithological and geomorphological information was used to inform the suitable depths to collect OSL samples (see below).

### 3.2 Analysis of historical maps

Maps show the position of the river channel at the time of the map survey, however the use of this information for quantitative reconstruction of channel position over time is not straightforward. First of all, historical maps are likely less accurate than their modern counterparts. Hence there is an uncertainty in the exact position of the channel. In this section we outline how we quantify this spatial uncertainty. Next, to allow Bayesian analysis of the sequence, this spatial uncertainty must be translated into a temporal uncertainty. The necessity of this stap and

the approach we take are outlined in section 3.4.

### 3.2.1 Georeferencing

Six historical maps (dating from 1720 to 1894/1901 AD, indicated with letters a-f, Table 1, Fig. 2a-f) were selected for analysis based on the level of detail displayed. Younger maps (example 1931 AD in Fig. 2g) did not provide

relevant information as the meanders under investigation were already cut-off during river normalization in the early twentieth century. Older maps (>1720 AD) were also available for the area, but had to be excluded from the analysis as georeferencing was impossible due to the limited topographical information on the maps. The selected maps vary in extent (e.g. focused on the Overijsselse Vecht or on The Netherlands as a whole) and were made for different purposes (military or topographical surveys), consequently they vary in detail and quality for the study

area. The relevant map parts were scanned or saved at a resolution of 600 dpi, or otherwise used with the given image quality. The maps were georeferenced using Esri's ArcMap (versions 10.3.1 and 10.5) based on static landscape features (e.g. churches, road crossings) that were displayed on both historical and reference maps. For the latter we used a digital elevation model (AHN2, 5 m horizontal resolution, 0.2 m vertical resolution) and a recent topographical map (OpenTopo) (AHN, 2018; Van Heerd and Van't Zand, 1999; Van Aalst, 2017). Finding

high-quality ground control points (GCPs) was challenging due to the nature of the maps. We predominantly made use of clearly delineated features such as road crossings, but occasionally included features with more irregular edges such as arable fields. Maps were georectified using a first order polynomial transformation (e.g. Downward et al., 1994; Leys and Werritty, 1999) and nearest neighbour pixel resampling as this resulted in the sharpest display of the river channel (based on visual assessment). The ground control points were exported from ArcMap

as a *.txt file to use in the freely available software MapAnalyst (see below). Map quality was analysed using (1) a quantification of geospatial error or planimetric accuracy and (2) with visualisations of geodetic distortions as generated by MapAnalyst. Both aspects are explained below.

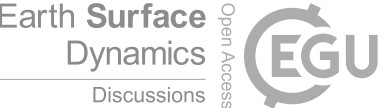



**Figure 2.** (a-e) Fragments of historical maps displaying the study area (for explanation refer to Table 1), (f) map showing the same area anno 1931 (publication date). As fragment (f) consists of two map sheets two dates are listed (details in Table 1). Image references: (a) Algemeen Rijks Archief. Genie-archief, situatiekaart 07; (b) Versfelt, 2003; (c) Versfelt and Schroor, 2005; (d-g) CC-BY Kadaster, 2018.

### 3.2.2 Quantifying geospatial error

The root mean square error (RMSE) of the displacement vectors of the GCPs is often used as a quantification of geospatial error and was automatically calculated by ArcMap (Esri, 2018) according to:

$$RMSE = \sqrt{\frac{1}{n}\left(\sum_{i=1}^{n}\left(\varepsilon_{x_i}^2 + \varepsilon_{y_i}^2\right)\right)} \qquad (1)$$

Where $\varepsilon_{x_i}$ and $\varepsilon_{y_i}$ respectively represent the x and y component of the displacement vector that indicates the difference between the location of GCP i on the reference map and the location of the same point on the historical map after first order polynomial transformation. The total number of GCPs is indicated by n. The RMSE was used to assess geospatial error or planimetric accuracy, defined as 'the extent to which distances and bearings between identifiable objects coincide with their true values' (cf. Jenny and Hurni, 2011).

After georeferencing the channel centreline was deduced from each map to reconstruct channel position for each time slice. The RMSE was used to draw an error band around each centreline. The centreline and error band represent the positional mean and related standard deviation respectively. The channel centrelines and error bands were subsequently used in the chronological modelling (see below). The final position of both investigated meanders was based on the abandoned channel as displayed by the DEM of the present-day situation. Historical evidence indicates that Prathoek was channelized between 1884 and 1901 AD (Fig. 2ef). Junner Koeland was channelized between 1894 (Fig. 2f) and 1906 AD, as indicated by the topographical map of the Netherlands 1:50.000, map sheet Coevorden, surveyed between 1900-1906 (not shown).

### 3.2.3 Visualisation of geodetic distortion

Distortion grids were generated for each map fragment based on the GCPs using the software MapAnalyst (Jenny and Hurni, 2011; http://mapanalyst.org/). A world file (*.txt) was created for the reference image and together with the historical map image loaded in MapAnalyst. In the *.txt file with GCPs that was generated with ArcMap the first two columns show the map points indicated in pixel rows/columns, using the upper left corner as origin. The third and fourth column show the points of the reference image, using the units of its coordinate system (in our case the Dutch RDnew projection with units in metres). Using a text editor the points for the map and reference image were split into two separate files. In addition each point pair was assigned an ID number in the first column to allow MapAnalyst to link the point pairs from the two files. For the map images MapAnalyst uses the lower left corner as origin instead of the upper left corner as is done in ArcMap, therefore the pixel values in y-direction for the map points were converted prior to loading them in MapAnalyst. Map distortions were calculated using a 1 kilometre raster and exported as *.png images.

### 3.3 OSL dating

Optically stimulated luminescence (OSL) dating makes use of a small light signal emitted by mineral grains (e.g. quartz) upon optical stimulation. The intensity of the signal can be linked to the age of the sediment. The (latent)



OSL signal builds up as the grains are exposed to natural ionizing radiation in the environment (from uranium, thorium, potassium and cosmic rays; known as the environmental dose), which causes accumulation of trapped charge in the crystal structure. The moment the grains are exposed to daylight during transport, the trapped charge is released, and the OSL signal is zeroed (or bleached). When the grains are deposited and buried, they are no longer exposed to light, and the signal can build up again.

Through comparison of the natural luminescence signal with that induced by laboratory irradiation, the environmental dose received by the grains since burial can be estimated (the palaeodose). Combined with information on the environmental dose rate experienced by the grains, the age can be calculated by: age [ka] = palaeodose [Gy] / dose rate [Gy/ka] (see e.g Rhodes, 2011; Jacobs, 2017 for more information on OSL dating).

With dating of fluvial sediments, the assumption in luminescence dating that the OSL signal was completely zeroed prior to sediment burial is questionable (Murray et al., 1995), due to the limited exposure of grains to light when transported in turbid water (e.g. Wallinga, 2002a). This means that at the time of sediment burial, some trapped charge that was build up since previous deposition may remain, which may consequently lead to a remnant OSL signal upon deposition, and an overestimation of the age. Poorly bleached sediments mostly consist of a mixture of poorly bleached and well-bleached grains (Olley et al., 1999), which can be detected by broad and skewed equivalent dose distributions (e.g. Wallinga, 2002b). Heterogeneous bleaching in channel site samples may result in apparent doses ranging over several orders of magnitude and may limit the accuracy of dates (Murray et al., 1995; Olley et al., 1998), although methods have been developed to largely overcome these challenges (e.g. Galbraith et al., 1999; Cunningham and Wallinga, 2012).

### 3.3.1 OSL sampling

In total 10 samples were collected for OSL dating to determine the time of deposition and burial of the sediments. Six samples were collected in the Junner Koeland point bar, three samples in the point bar of Prathoek, and one sample was collected in an Early-Holocene fluvial terrace remnant near Prathoek. The point bar samples were collected just above the channel lag deposit, because (1) the channel lag can generally easily be recognized, thereby preventing accidentally sampling older deposits below the point bar deposit, (2) at this depth recent aeolian reworking of sediment can be excluded, and (3) the channel lag is located below the oxidation-reduction zone, backing the assumption that the sediments have been water saturated since deposition (and thus reducing uncertainty in calculating the environmental dose rate).

Samples for OSL dating were obtained through a modified Van der Staay suction corer (Wallinga and Van der Staay, 1999). First a hole was augered down to the desired depth, using an Edelman auger above the groundwater level and the suction corer below groundwater level. Then, the suction corer, with a loosely fitted PVC sample tube of 30 cm length at its end, was inserted in the hole and pushed down to the lower sample depth. After retrieving the suction corer, the sample tube was removed, and both ends were immediately covered with plastic caps and taped with light-impermeable black tape. Sampling locations were recorded with a GPS device with a horizontal precision of around 3 m.

### 3.3.2. OSL measurements

The samples were analysed in the laboratory of the Netherlands Centre for Luminescence dating in Wageningen, The Netherlands. Under amber safelight conditions, the sample tubes were opened and split in two fractions.



Material from the light-exposed outer ends of the tube was prepared for dose rate estimation (section 3.3.3), while the material from the inside of the tube was suitable for palaeodose estimation. Luminescence measurements were performed on quartz grains of the fraction 212 - 250 µm. This fraction was obtained by a combination sieving and subsequent treatment with HCl (to remove carbonates), $H_2O_2$ (to remove organic material), and HF (to dissolve

feldspars and etch the quartz grains). To analyse the samples the Single Aliquot Regenerative-dose protocol (SAR) (Murray and Roberts, 1998; Murray and Wintle, 2000, 2003) was used, with details as listed in Table 2. The preheat temperatures were chosen based on pre-heat plateau tests. With the applied procedure a laboratory dose (i.e. exposure to a known amount of radiation) could accurately be recovered, as indicated by the average dose recovery ratio of $1.03 \pm 0.01$. At least 22 aliquots each consisting of ~75 grains (2 mm mask size) were measured

per sample (31 aliquots on average). The equivalent dose distributions were displayed in radial plots, constructed with the Luminescence package (Kreutzer et al., 2015) employed in R (version 3.2.4).

### 3.3.3. Dose rate

Light-exposed material from the outer ends of the sample tubes was prepared for dose rate estimation. Samples

were dried and combusted, allowing measurement of water and organic content, and then mixed with wax and moulded into pucks. These were measured on a high resolution broad range gamma-ray spectrometer, to determine radionuclide concentrations of K-40 and several radionuclides in the U and Th decay chains. From these, dose rates experienced by the quartz grains were calculated, taking into account attenuation due to water, organics, and grainsize, and adding a contribution from cosmic rays (assuming instant burial to present depth). Methods are

detailed in Wallinga and Bos (2010).

### 3.3.4 Palaeodose estimation

We analysed the equivalent dose distributions of all samples from point bar deposits using the unlogged version of the bootstrapped Minimum Age Model (bsMAM; Galbraith et al., 1999; Cunningham and Wallinga, 2012).

This approach is specifically suited for this project as it takes into account uncertainty on sigmab (see below), and allows the construction of likelihood distributions of the palaeodose, which can be used as priors for Bayesian analysis (Cunningham and Wallinga, 2012). The unlogged version of the MAM was used, as the logged version was not applicable to the youngest samples. Being aware of reservations with regard to using the unlogged model to older samples (Arnold et al., 2009), we checked that for older samples results of logged and unlogged models

were in agreement.

The (bootstrapped) MAM requires an estimate of overdispersion in the distribution that is not caused by heterogeneous bleaching (sigmab). Ideally, this estimate is based on the observed overdispersion on well-bleached samples from the same lithology, depositional environment and age (e.g. Chamberlain et al., 2018). As none of the samples obtained from the Holocene point bars appeared to be well bleached, we had to take another approach.

This alternative is provided by the overdispersion obtained on the oldest and best-bleached sample (NCL-2415163), which was collected from an Early-Holocene terrace remnant. Although the depositional environment and age are different, the lithology of the material is nearly identical as the point-bar deposits consists mostly of reworked fluvial deposits of Late Pleistocene/Early Holocene age. The overdispersion of sample NCL-2415163 was $0.15 \pm 0.03$ and this value was used as sigmab input for the bsMAM model. The bootstrapped MAM analysis

of the equivalent dose distribution is combined with the sample dose rate to provide a likelihood distribution of



estimated burial ages for each sample (see Cunningham and Wallinga, 2012). These age likelihood distributions are implemented as OSL priors in the Bayesian modelling (see 3.4). Sample NCL-2415163 did not suffer from poor bleaching and based on the normally distributed and relatively narrow equivalent dose distribution for this sample, the Central Age Model (CAM, Galbraith et al., 1999) was adopted for palaeodose estimation.


### 3.4 Bayesian reconstruction of river channel position

Bayesian statistical approaches were used to combine the numerical age data obtained from the historical maps and OSL dating, with auxiliary data on the stratigraphic order. This analysis was performed using the freely available OxCal software (version 3.4, first described by Bronk Ramsey, 1995). Bayesian analyses of deposition

sequences are normally based on age versus depth profiles (Bronk Ramsey, 2008). Here we analyse age as a function of lateral migration distance, where the location of the oldest swale (meander base) is taken as origin, and distance is measured along the central axis of the scroll bars. For each of the age constraints (OSL sample age or survey age of a historical map), we have both an age and a location along the central axis. In addition, we know that the meander grew from its base to the final position, providing an order for all information. In the Bayesian

approach, all this information is combined to obtain a robust reconstruction of the channel position through time.

To jointly analyse the OSL priors and the historical map data, both were added as entries in a Bayesian deposition model using the P-Sequence (Fig. 3). This command fixes the chronology of the sampling points using the known order of events (based on the known meander expansion direction) and makes use of a Poisson distribution. With this depositional model, channel migration is assumed to be random (but always positive) giving

approximate proportionality to distance (factor $z$) (cf. Bronk Ramsey, 2008). The P-sequence model needs information on the size of the units (deposition events) within the sequence. Here we chose to use the average dimension of a point bar as unit, so the mean distance between two swales. Based on meander amplitude and the visual counting of point bars from the DEM, this distance was estimated at 77 and 42 m, for Junner Koeland and Prathoek respectively. This provides an event spacing ($k_0$-value) of 0.013 events m$^{-1}$ for Junner Koeland and 0.024

events m$^{-1}$ for Prathoek. By applying the regular Boundary function the entire distributions were allowed to be used in the calculation.



```
Plot()
 {
  P_Sequence("Junner Koeland",.013)
  {
  Boundary("Meander base", Date()){ z=1190; };
  Prior("NCL2415159", NCL2415159){ z=1001; };
  Prior("NCL2415158", NCL2415158){ z=856; };
  Prior("NCL2415157", NCL2415157){ z=591; };
  Date("EO1829"){ z=361; };
  Date("EO1851"){ z=356; };
  Date(N(AD(1851),14)){ z=311; };
  Date(N(AD(1829),16)){ z=302; };
  Date("EY1851"){ z=266; };
  Date("EY1829"){ z=256; };
  Prior("NCL2415155", NCL2415155){ z=233; };
  Date("EO1884"){ z=182; };
  Date(N(AD(1884),6)){ z=141; };
  Date("EY1884"){ z=99; };
  Prior("NCL2415154", NCL2415154){ z=6; };
  Date(U(AD(1894),AD(1906))){ z=0.01; };
  Boundary("Meander apex", Date()){ z=0; };
  };
 };
```

**Figure 3.** OxCal script as used for the fourth iteration for Junner Koeland. A similar script was used for Prathoek.


Likelihood distributions of estimated burial age (see section 3.3.4) were uploaded to the OxCal directory as priors (*.prior file), with the unit in years AD. For this OSL age data, the position is exactly known, while there is uncertainty in the time domain (age estimate).

For the historical map data the age is known exactly, while reconstructed locations are uncertain. Incorporating this data in the Bayesian model requires uncertainty to be expressed in the time domain, rather than spatial domain. Moreover, a slight correction is needed as the OSL data provide age information on the point bar formation (i.e. inner bank), while the channel centreline was reconstructed from the maps. Channel width varied significantly between the historical maps and may not represent the true channel width of past times. The historical river channel was therefore deduced from each map as a channel centreline. To make sure that the age of the map is assigned to the inner bank instead of the channel centre in the chronological analysis we corrected the distance ($z$-value) for each centreline using half the channel width. For this calculation, we assumed that channel width has been relatively stable throughout meander formation, and is represented by the width of the abandoned channel ($\pm$ 30 metres).

Uncertainty in the spatial domain was calculated for each map based on sigma (see 3.2.2) and assuming a normal distribution. To transfer this spatial uncertainty in an age uncertainty, the migration rate of the channel is required. This provides a problem, as the procedure aims at a reconstruction of channel position through time, which is needed to determine migration rate. Nevertheless, a crude estimation of migration rate is possible based on available data, and an iterative approach can be used to improve this estimation with the model output. For the first iteration error bands were converted to years using an estimated migration rate based on the bsMAM age of the OSL sample closest to the meander base and historically known maximum channelisation date (1894 and 1906 for Prathoek and Junner Koeland respectively), when the channel reached its final position. For subsequent iterations migration speed was calculated for each map individually by dividing the difference between the modelled mean ages for its error boundaries (in years) by the error width (in m). This migration rate was then used to convert RMSE to years. The converted RMSE (i.e. sigma) was used together with the map age (i.e. the mean)



as input for the next iteration. This process was repeated until model outcome converged and remained stable. This
       required four iterations of the Bayesian model for both investigated meander bends.

       Queries were built into the model to generate output for requested distances, including the meander base
       (calculated by OxCal using extrapolation of the model, consequently the uncertainty for this point will be larger)
       and the spatial boundaries of each error band to allow calculation of a migration speed for each historical map as
described above.

       Upon convergence and stabilisation of the model outcome the deposition model of the final (i.e. fourth)
       iteration was used to interpolate ages throughout the point bar. Using the graphical options in OxCal interpolation
       results were plotted versus distance from the meander base as origin toward the meander apex using linear
       interpolation in between the data points. Plots of the input data were created with the ggplot2 package (Wickham
and Chang, 2016) employed in R (version 3.4.0).

       We performed sensitivity tests to check model sensitivity to (1) the initial migration rate that was used to
       convert error bands to years, (2) the correction for channel width and (3) a rigid versus variable $k_0$-value. Test
       outcomes indicated that (1) either increasing or decreasing the initial migration rate by a factor of two yielded
       similar results after several iterations, (2) model outcome with and without using the channel correction differed
only a couple of years, and (3) a rigid or variable $k_0$-value resulted in age differences of no more than one year.
       Based on these tests we concluded that model sensitivity to these factors is minor.

## 4 Results

### 4.1 Historical maps

Figure 4 shows the distortion grid for the map of 1851 AD. As the grid appears virtually straight, geodetic
distortions in the map fragment are minimal. Distortion grids for the other maps were similar (provided in the
supplement).

       The channel centrelines that were deduced from the maps are shown in Fig. 5. On the map of 1894 Prathoek is
already channelized (Fig. 2), therefore this line is only drawn for Junner Koeland. Based on the current DEM and
       coring evidence, the georeferenced map dating from 1720 had to be rejected from the Bayesian modelling based
       on obvious erroneous representation of the river course. The 1720 map depicted the Junner Koeland meander
       where corings proved that river deposits were lacking, i.e. too far west by up to 80 metres. We tried to include the
       georeferenced map of 1786 in the Bayesian modelling, but this map resulted in poor model agreement (further
discussed below). The correctness of this map was doubted, because it displays the meander apex at Prathoek
       directed southwards instead of southwest like the other channel centrelines (Fig. 5). This would indicate that
       sedimentation towards the inner bend has taken place in order to move the apex westward, which seems unlikely.
       Based on this improbable meander planform and the statistical issue created by this map in the Bayesian modelling
       we excluded it from further analyses.




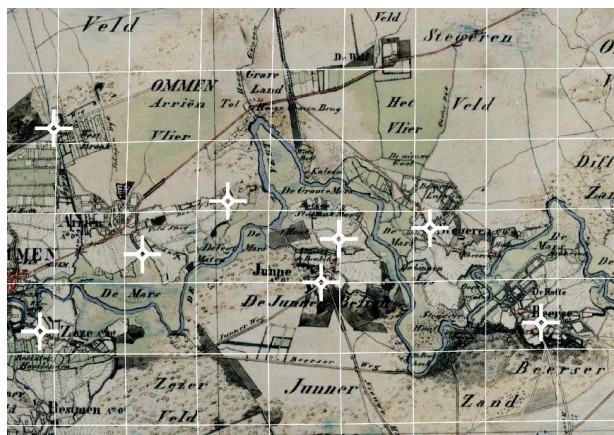

**Figure 4.** Fragment of the georeferenced historical map of 1851 AD (details listed in Table 2), displaying ground control points (white plusses) and a distortion grid (in white, maze size of 1 km). The grid approaches a perfect grid, indicating that map distortion is minimal. Grid generated with MapAnalyst software by Jenny and Hurni (2011). Image reference historical map: CC-BY Kadaster, 2018.

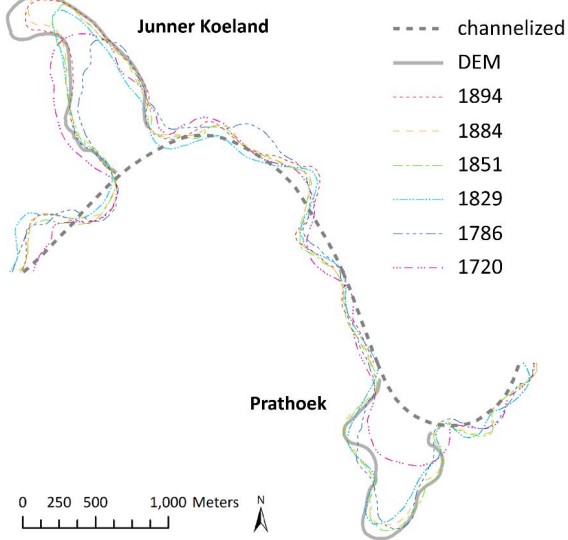

**Figure 5.** Channel centrelines as derived from the historical maps (years AD) listed in Table 1 and shown in Fig. 2. The DEM (digital elevation model) line represents the course of the meanders preserved upon abandonment at the time of channelization (between 1884-1901 AD for Prathoek and between 1894-1906 AD for Junner Koeland).

### 4.2 OSL ages

The results of the OSL dating and bsMAM calculations are presented in Table 3. The spread in the equivalent doses of single aliquots for the point bar samples (example in Fig. 6a) shows that light exposure of the grains prior to deposition and burial was not sufficient to completely reset the quartz OSL signal of all grains. Strikingly the



equivalent dose distribution of many samples shows two distinct peaks; a younger population representing the age of recent transportation and deposition by the river, and an older population with an age of about 11 ka, likely representing the depositional age of the grains prior to erosion and transport of the Late-Holocene Vecht river. It is noteworthy that the sample collected in a terrace remnant at Prathoek was also dated at circa 11 ka (Fig. 6b), 390 corroborating the assumption that the second peak in the distribution represents an earlier depositional event.

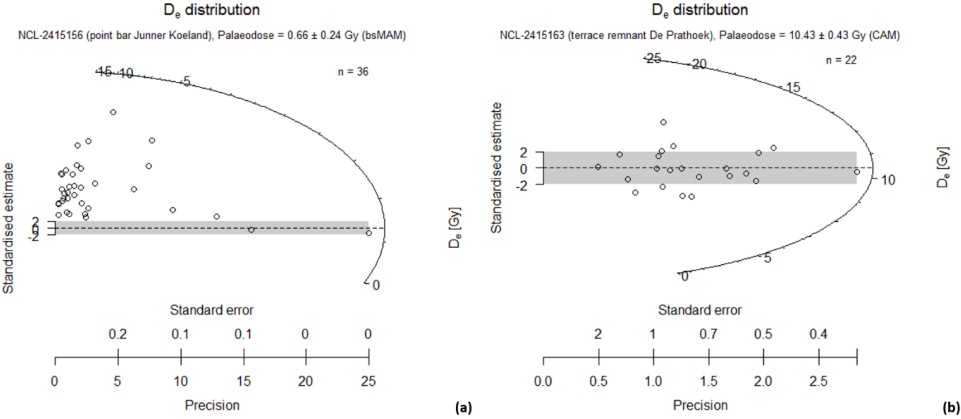

**Figure 6.** Two radial plot examples showing (a) a point bar sample (NCL-2415156) from Junner Koeland, and (b) the sample collected in a terrace remnant near Prathoek (NCL-2415163). The point bar sample in (a) is poorly bleached, showing a large 395 population of grains with a $D_e$ of about 10 Gy. Such equivalent doses are similar to those of the terrace sample in (b) (CAM: $D_e$ 10.43 ± 0.43 Gy).

### 4.3 Reconstructed channel position and chronology

The Bayesian model was iteratively repeated until model results converged to similar output, which occurred after 400 four iterations for both Junner Koeland and Prathoek, therefore the deposition models that resulted from these iterations were used to interpolate ages throughout the point bar. Table 4 lists both the unmodelled and modelled ages. The model agreement index (A-value) should be above 60% to indicate proper model functioning (Bronk Ramsey, 1995). The high A-values in Table 4 show that the modelled ages fit the data well. The graphical output of the model as generated by OxCal is displayed at the right in Fig. 7bd and demonstrates the improvement of the 405 geochronology after the Bayesian modelling compared to the unmodelled data points shown at the left in Fig. 7ac.

OSL sample NCL-2415156 suffered from very poor bleaching (Fig. 6a) and therefore had to be rejected from the Junner Koeland model, otherwise the model A-value did not rise above 60%. In addition, the historical map of 1894 had to excluded, as it resulted in poor agreement that could not be solved by adjusting the model (e.g. $k_0$-value). In the fourth (and final) iteration the model had an A-value of 143.8% (Table 4a). For Prathoek the poorly 410 bleached sample NCL-2415160 resulted in poor agreement; after excluding this entry the A-value of the fourth iteration amounted to 112.2%.

The Bayesian model also allows extrapolation of the results towards the meander base (Fig. 7). In our study area the positions of the two meander bases appear to be located close to the former valley edge, therefore the modelled ages of the meander bases indicate the moment when the meanders sufficiently eroded the valley edge 415 to break free from lateral confinement. Model results indicate that this took place during the fourteenth century AD for Prathoek and about 65 years later during the fifteenth century for Junner Koeland. Subsequently the





meanders migrated outside the river valley at an average rate of 2.6 and 0.9 m/y for Junner Koeland and Prathoek respectively (calculated by dividing the distance from meander base to apex by the difference between their modelled mean ages). For both meanders, there seems to be an increase in lateral migration rate in the final stages

prior to abandonment; roughly between 1825 and 1900 AD.

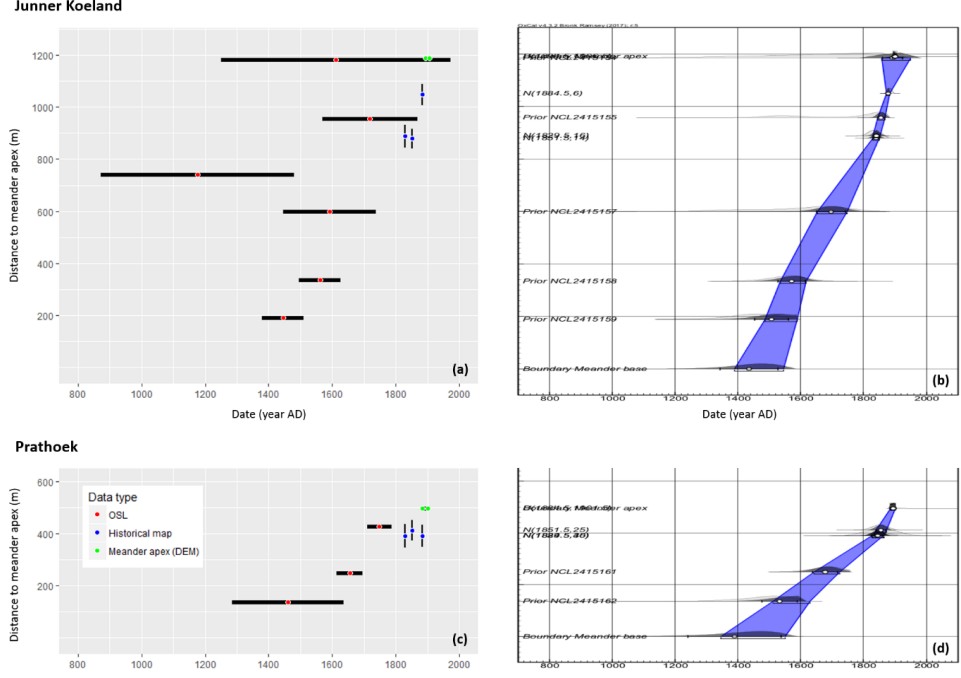

**Figure 7.** Comparison of results before (a, c) and after (b, d) the Bayesian modelling for Junner Koeland (a, b) and Prathoek (c, d). Both (a, b) and (c, d) have equal y-axes. The legend in (c) also applies to (a). In (a, c) the historical map data are indicated
with error bars displaying RMSE of the georeferencing procedure (in m), the OSL results of the bootstrapped Minimum Age Model indicate the modelled mean and 1 sigma error bar (in years). In (b, d) the blue areas in the Bayesian models indicate the 68.2 % confidence interval. The data point indicating the meander base in (b, d) is calculated through extrapolation by the Bayesian model.

**5 Discussion**

Our study demonstrates that the presented Bayesian chronological modelling approach combining OSL and historical map data improves the robustness of fluvial chronologies and can elongate them into longer timeframes, compared to what could be achieved by using either data type separately (Fig. 7).

The use of historical maps to determine fluvial migration rates is often challenging due to limited options for
georeferencing and uncertainty regarding planimetric accuracy. Historical maps can offer a diachronic perspective on the development of meander shape and channel position (Fig. 2, 5). For our case study this was however limited to recent history (after 1720 AD). When going further back in time, the availability of historical maps was limited and map quality was generally too coarse for a detailed reconstruction of meander migration. Doubts regarding the correctness of the maps dating from 1720 and 1786 resulted in excluding these maps from the Bayesian analysis
in our study.





With a limited number of GCPs RMSE might underestimate the geospatial error (Esri, 2018; Hughes et al., 2006). When used as a rigid buffer around channel centrelines to determine whether channel changes larger than the assumed error buffer occurred (i.e. deviations that are so large that they cannot emerge from error) (e.g Urban and Rhoads, 2003; Rhoades et al., 2009), this may result in not detecting small channel changes and could lead to
false interpretations of channel stability. Other approaches have been proposed to account for this problem by analysing independent test points or using cross-validation techniques to develop spatially variable error buffers (Hughes et al., 2006; Lea and Legleiter, 2016). However, these methods were developed for aerial imagery where temporal uncertainty is absent and do not convert uncertainty in the spatial domain to uncertainty in the temporal domain. Additionally these methods generally require higher numbers of GCPs than could be registered for
georeferencing the historical maps (e.g. 35 per image as used by Lea and Legleiter, 2016). In contrast, the Bayesian modelling methodology that we have presented permits a flexible interpretation of the RMSE buffer, which can easily be calculated even with lower numbers of GCPs, and the iterative procedure with the Bayesian model allows conversion of the geospatial error into temporal uncertainty. This temporal uncertainty is included in the model as a probability density function, i.e. allowing all channel positions but each with a specific probability of occurrence.

The use of OSL to date young fluvial sediments is challenging, as limited light exposure during subaquaous transport of grains in a turbid river may lead to incomplete resetting of the OSL signal in at least part of the grains (Wallinga, 2002a). As demonstrated by Fig. 6 the sediments were indeed affected by poor bleaching, in some cases such extreme that the equivalent dose determined on most aliquots reflected the burial dose of the source material, rather than the deposit of interest. The bootstrapped MAM approach yields very large uncertainties in such cases,
and the Bayesian analysis indicates that these uncertainty estimates are reasonable. Only for two samples (NCL-2415156 and NCL-2415160, i.e. one from each meander) was the OSL age rejected by the Bayesian model. The Bayesian modelling significantly enhanced the interpretation of the OSL dating results and the robustness of the geochronology.

Through extrapolation the Bayesian model could estimate the date of meander expansion beyond the valley
edges. From this pivotal moment in time, the meanders were no longer confined and large meanders formed. Our results indicate average lateral migration rates of 2.6 and 0.9 m/y for Junner Koeland and Prathoek respectively. It seems likely that migration rates may have been much higher during brief periods, alternating with periods of relative standstill (Gurnell et al., 1994). However, our chronological data lacks the precision and resolution to identify such fluctuations, possibly with the exception of a period with rapid migration in the period 1825-1900
AD.

The methods outlined in this paper provide the chronological constraints that are needed to relate meander expansion beyond the valley edge and changing river dynamics to climate and land use changes. Such analysis is beyond the scope of this contribution, and reported separately by Candel et al. (submitted).

**6 Conclusion**

Our study demonstrates that combining historical map data and OSL dates in the presented Bayesian modelling approach yields an integrated geochronology with a higher robustness than the sum of its parts. The analysis requires conversion of geospatial error from the historical maps into a temporal uncertainty, and we show that this is possible through an iterative approach. Our method incorporates strengths of both data types (historical maps
and OSL dating) for reconstructing fluvial morphodynamics and supports elongating fluvial sequences into larger



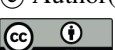

time frames. Our study indicates that the meanders of the Overijsselse Vecht expanded beyond the valley margins from about 1400 AD onwards, and that the channels migrated at an average rate of up to 2.6 m/year during their lifetime (1400 – 1900 AD).

**7 Competing interests**

The authors declare that they have no conflict of interest.

**8 Acknowledgements**

We thank Ruud Jonker (Staatsbosbeheer Vechtdal) and Jan de Roos (Camping de Roos) for the permission to do
field work at Junner Koeland and Prathoek; Jasper Candel and Wim Hoek for field assistance with OSL sample collection; Alice Versendaal and Erna Voskuilen of the Netherlands Centre for Luminescence dating for their work on the OSL samples; Gerard Heuvelink and Sytze de Bruin for discussions on planimetric accuracy issues and statistics; Roy van Beek, Erik van den Berg and Gilbert Maas for assistance in collecting historical maps; students Sjoukje de Lange, Jip Zinsmeister, Pascal Born and Karianne van der Werf of Utrecht University for their work
on the point bar transects; and Duco de Vries for his explanation on the R Luminescence package.

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

morfodynamiek en kansrijkdom voor natuurontwikkeling. Wageningen.





## 10 Tables

**Table 1.** Overview of the six historical maps used in this study. Revision date indicates full or partial revision. The date used for analysis is based on the revision date, or if none, on the survey date. For the map sheet of the Hottinger atlas the survey covered three years, we therefore took the middle year to use in further analyses. GCPs = Ground Control Points (as used for georeferencing), RMSE = root mean square error (expressed in metres) based on the residuals (or displacement vectors) of the GCPs after using a first order polynomial transformation in the georectification procedure. JK = Junner Koeland, PH = Prathoek. Map sections are shown in Figure 2. References are detailed in the subscript.

| Name | Original scale | Map sheet | Survey date | Revision date | Publication Date | Date used for analyses | No. GCPs | RMSE (m) | Figure 2 | Reference |
|---|---|---|---|---|---|---|---|---|---|---|
| 'Limiten tussen Bentheim en Overijssel' by Pieter de la Rive | 1:19200 | - | unknown | unknown | ± 1720 | 1720 | 8 | 104.482 | a | 1, 2 |
| De Hottinger-atlas van Noord- en Oost-Nederland | 1:14400 | 35, 36 | 1785 – 1787 | none | 1773-1794 | 1786 | 39 | 106.702 | b | 3 |
| Atlas of Huguenin | 1:40000 | 72, 73 | 1829 | none | 1829 | 1829 | 15 | 45.229 | c | 4 |
| Topographische en Militaire Kaart van het Koninkrijk der Nederlanden | 1:50000 | 22 | 1851 | none | 1859 | 1851 | 11 | 39.056 | d | 5, 6 |
| Topographical map of the Netherlands (Bonne) [i] | unknown | Coevorden | unknown | 1884 | 1897 | 1884 | 14 | 41.038 | e | 5 |
| Topographical map of the Netherlands (Bonne) [ii] | 1:25000 | 306 (JK) 307 (PH) | 1883 1901 | 1894 none | 1896 1904 | 1894 1901 | 10 | 27.073 | f | 5 |

1 Algemeen Rijks Archief. Genie-archief, situatiekaart 07.
2 Wolfert et al., 1996.
3 Versfelt, 2003.
4 Versfelt and Schroor, 2005.
5 CC-BY Kadaster, 2018.
6 Van der Linden, 1973.

**Table 2.** The SAR protocol used for equivalent dose measurement, including measurement parameters.

| Step | Action | Measured |
|---|---|---|
| 1 | Beta dose (or Natural dose) | |
| 2 | 10s preheat at 200 °C | |
| 3 | 20s blue stimulation at 125 °C | $L_n$, $L_i$ |
| 4 | Beta test dose | |
| 5 | 10s cutheat to 180 °C | |
| 6 | 20s blue stimulation at 125 °C | $T_n$, $T_i$ |
| 7 | 40s blue bleach at 210 °C | |
| 8 | Repeat step 1-8 for a regenerative dose, zero and repeat dose | |
| Extra 1 | Repeat step 1-8 with added infrared bleach at 30 °C prior to step 4 | |





**Table 3.** Results of the OSL dating. JK = Junner Koeland, PH = Prathoek, RD = Dutch coordinate system (Rijksdriehoekstelsel). Palaeodoses and ages are based on the bootstrapped Minimum Age Model for all samples from point bar deposits (JK and PH), and the Central Age Model for the terrace sample.

| Sample | | Location (RD coordinates) | | Sample depth below surface (m) | | Palaeodose (Gy) | | Total dose rate (Gy/ka) | | OSL age (ka) | | OSL age (year AD) | |
|---|---|---|---|---|---|---|---|---|---|---|---|---|---|
| Code | Site | x | y | Upper limit | Lower limit | μ | σ | μ | σ | μ | σ | μ | σ |
| NCL-2415154 | JK | 228508 | 506133 | 1.9 | 2.2 | 0.34 | 0.30 | 0.83 | 0.03 | 0.40 | 0.36 | 1612 | 362 |
| NCL-2415155 | JK | 228718 | 506204 | 1.2 | 1.5 | 0.26 | 0.13 | 0.87 | 0.03 | 0.30 | 0.15 | 1719 | 149 |
| NCL-2415156 | JK | 228869 | 506095 | 2.4 | 2.7 | 0.66 | 0.24 | 0.79 | 0.03 | 0.84 | 0.30 | 1177 | 304 |
| NCL-2415157 | JK | 229018 | 506021 | 2.4 | 2.7 | 0.37 | 0.13 | 0.86 | 0.03 | 0.42 | 0.15 | 1591 | 146 |
| NCL-2415158 | JK | 229108 | 505779 | 2.3 | 2.5 | 0.42 | 0.06 | 0.93 | 0.03 | 0.46 | 0.07 | 1561 | 66 |
| NCL-2415159 | JK | 229183 | 505655 | 2.8 | 3.1 | 0.47 | 0.05 | 0.82 | 0.03 | 0.57 | 0.07 | 1445 | 65 |
| NCL-2415160 | PH | 231070 | 502684 | 2.2 | 2.5 | 0.28 | 0.04 | 1.04 | 0.04 | 0.27 | 0.04 | 1749 | 38 |
| NCL-2415161 | PH | 231113 | 502848 | 1.6 | 1.9 | 0.34 | 0.04 | 0.94 | 0.03 | 0.36 | 0.04 | 1655 | 42 |
| NCL-2415162 | PH | 231122 | 502965 | 2.8 | 3.1 | 0.46 | 0.15 | 0.83 | 0.03 | 0.56 | 0.18 | 1461 | 176 |
| NCL-2415163 | terrace | 231167 | 503366 | 3.3 | 3.6 | 10.43 | 0.43 | 0.95 | 0.03 | 10.9 | 0.6 | NA | NA |



**Table 4.** Results of the Bayesian model for (a) Junner Koeland, iteration 4 and (b) Prathoek, iteration 4.

| (a) | Unmodelled (AD) | | Modelled (AD) | | Indices |
|---|---|---|---|---|---|
| | | | | | $A_{model}$ = 143.8 |
| | | | | | $A_{overall}$ = 149 |
| Name | μ | σ | μ | σ | A |
| P_Sequence Junner Koeland | | | | | |
| Boundary Meander base | | | 1433 | 92 | |
| Prior NCL2415159 | 1443 | 79 | 1506 | 54 | 97.7 |
| Prior NCL2415158 | 1555 | 65 | 1570 | 43 | 116.1 |
| Prior NCL2415157 | 1576 | 165 | 1695 | 46 | 143.2 |
| EO1829 | | | 1813 | 29 | |
| EO1851 | | | 1816 | 28 | |
| N(1851.5,14) | 1851 | 14 | 1839 | 10 | 91.8 |
| N(1829.5,16) | 1829 | 16 | 1840 | 10 | 100 |
| EY1851 | | | 1848 | 11 | |
| EY1829 | | | 1850 | 12 | |
| Prior NCL2415155 | 1709 | 144 | 1855 | 12 | 177.4 |
| EO1884 | | | 1867 | 10 | |
| N(1884.5,6) | 1884 | 6 | 1877 | 4 | 77.6 |
| EY1884 | | | 1879 | 4 | |
| Prior NCL2415154 | 1515 | 430 | 1892 | 32 | 161 |
| U(1894.5,1906.5) | 1900 | 3 | 1899 | 3 | 100 |
| Boundary Meander apex | | | 1900 | 7 | |

| (b) | Unmodelled (AD) | | Modelled (AD) | | Indices |
|---|---|---|---|---|---|
| | | | | | $A_{model}$ = 112.2 |
| | | | | | $A_{overall}$ = 114 |
| Name | μ | σ | μ | σ | A |
| P_Sequence Prathoek | | | | | |
| Boundary Meander base | | | 1366 | 164 | |
| Prior NCL2415162 | 1431 | 211 | 1536 | 64 | 115.2 |
| Prior NCL2415161 | 1672 | 49 | 1676 | 41 | 107.4 |
| EO1829 | | | 1764 | 47 | |
| EO1884 | | | 1794 | 42 | |
| EO1851 | | | 1820 | 33 | |
| N(1829.5,40) | 1829 | 40 | 1843 | 18 | 122.5 |
| N(1884.5,35) | 1884 | 35 | 1844 | 17 | 76.6 |
| N(1851.5,25) | 1851 | 25 | 1854 | 16 | 118.8 |
| EY1884 | | | 1863 | 16 | |
| EY1829 | | | 1868 | 16 | |
| EY1851 | | | 1873 | 15 | |
| U(1884.5,1901.5) | 1892 | 5 | 1893 | 5 | 100 |
| Boundary Meander apex | | | 1894 | 10 | |

725