# Peer review of "Reconstructing lateral migration rates in meandering systems; a novel Bayesian approach combining OSL dating and historical maps"

_Earth Surface Dynamics, 2018_

## Referee Comment (RC1) · E. J. Rhodes (Referee) · 19 May 2018

Overall, this is an excellent contribution that is innovative, and has the potential to add significantly to the successful estimation of age and position (and therefore migration rates) in fluvial channels over the late Holocene. This is of great importance both for the academic community interested in understanding details of past environmental change, fluvial channel migration patterns and hydrological response to external factors, but also for those charged with understanding river migration from a flood management and engineering standpoint.

[Figure]

As part of this approach, the paper provides a practical solution for integrating two very different sorts of constraints for fluvial system evolution. These are: 1) map data where the primary uncertainty is spatial accuracy related both to survey precision and changes in channel width or pattern, but with a well established date, and 2) young OSL age estimates, where the sampling position is known with very high accuracy, but problems of incomplete signal bleaching lead to large uncertainty distributions in age.

This solution neatly transfers the spatial uncertainty into a time-uncertainty within a Bayesian age model, using an iterative approach. This takes advantage of the fact that the meander scroll bars develop in a single direction, allowing the application of a spatial sequence within the Bayesian age model. This approach substitutes the lithostratigraphic relative age control often used in Bayesian geochronological models of samples in a single vertical sequence, with a morphostratigraphic sequence provided by the scroll bar morphology and relationship to the modern channel. Note that in the cases studied here, the end point of this lateral migration is represented by channels abandoned when the river underwent significant engineering work around the start of the twentieth century.

Similar spatial morphostratigraphic Bayesian age models have been employed in other contexts (such as sequences of river terraces, or the development of beach ridges), but in my opinion, the approach developed and presented here is particularly appealing and well described, and the manner of building in the spatial uncertainty for the river channels to the age model is clever and effective.

I have a long list of specific points that refer to locations within the paper. But note, these are designed as a conversation or debate, and I hope will help to make the paper clearer and more accessible, and represent suggestions for the authors. The paper is enjoyable to read, and the maps are highly appealing, and are very reproduced to a high standard.

Line 41 It is probably more appropriate to say "humans" rather than "man" here.

Line 42 I suggest you add "and river management" after "land use". I am thinking of things such as weirs, mills, cut-off channels, bank modification etc.

Line 53 There are some other papers that do this, e.g. including Kemp and Rhodes (2010) QSR, though I am not suggesting you include the citation.

Line 70 I very firmly agree that this is underexplored.

Line 100 I would use a comma instead of a dot to read "50,000" and similar in other locations including Table 1.

Line 118 Possibly add "multiple" before "hand-corings" to help clarify the sampling strategy.

Line 120 Replace "on" with "of".

Line 127 "inform" is slightly awkward in this context; I understand your meaning, but it is usual to "inform someone of something…" so I suggest using "select" here.

Line 134 "Replace "stap" with "step".

Line 138 This is an excellent selection of maps.

Line 140 Is "normalization" a recognised term here? I would possibly suggest "regularization" (not great) or perhaps simply "channel management works" (better).

Line 145 It is not clear to me what "otherwise used" includes.

Lines 147-152 It seems sensible to refer to the supplement here.

Line 175 It would be informative to know the values of what the range of spatial errors was here.

Line 199 "small" for the OSL emission seems wrong to me. I think of small as meaning size, not intensity. I would say "low intensity" instead.

Line 203 "The moment…" is not 100% accurate, as some electrons are emitted and

recombine shortly after the light source is turned on. I would simply say "When" which has a lesser degree of suggestion of an instantaneous effect.

Line 204 As partial and incomplete bleaching are a feature of some samples, I would try and be more precise here. I suggest you say something like "the OSL signal is reduced (bleached) to a low level, often close to zero".

Line 213 Replace "build" with "built".

Line 225 The wording of "...generally easily be recognised..." is a little awkward. I suggest "...generally be recognised easily...

Line 228 I suggest replacing "backing" with "supporting".

Line 240 "...split in two fractions" does not quite work. "...split into two fractions" is better in terms of grammar, but is awkward to read and say, so consider rewording?

Line 244 It is useful to include the concentration and time of treatment for the HF here.

Line 260. We need a reference to the measurement conditions, which are presented in Table 2. It is useful to describe the equipment used and the filters used (probably U340 on a Riso set, but let's be sure). Note, in Table 2, "10s cutheat to 180 deg. C" doesn't make full sense. A cutheat by definition is for held for no time. I appreciate that the heat treatment in the second part of the SAR cycle is often referred to as a cutheat to distinguish it from the preheat before the main OSL measurement, so perhaps you could either just use "10s preheat at 180" here, or if you want to preserve cutheat, say "10s "cutheat" at 180..."?

Line 272 Replace "on" with "for" or "of".

Line 282 Refer to Fig. 6b, which includes this distribution.

Line 322 The use of +/- to mean "approximately" is a little misleading, and not recognised scientific usage. Either use the "∼" symbol, say "approximately". I was misled when I read this, as I assumed this was an uncertainty value on the width, rather than

the width value used. In fact, as you used this width value, there is not really any un-certainty on the value, only on whether it is appropriate (which I think it is!), so I would just say 30m here.

Line 335 It would be useful to know how much uncertainty in position there was here. Give us some typical or max/min values possibly.

Line 373 I would call these white symbols "crosses"; I don't think "plusses" is a recog-nised term.

Lines 399-401. Some of this information is repeated from a few line above, so you can contract this first sentence a little.

Lines 404, 405 The shorthand "Fig. 7bd" and "7ac" need writing out correctly.

Line 408 Is "resolved" better than "solved"?

Lines 440-450 Is it possible that the channel widths varied significantly, for example during the progression downstream of a gavel slug?

Table 1 Add a comma into the large numbers in the map scales, to read 1:19,200 etc. Replace +/- with $\sim$

Table 3 and elsewhere. Did you consider using CE (Common Era) rather than AD?

---

## Short Comment (SC1) · 1 Jun 2018

This is an interesting paper and potentially valuable. It is a commendable approach to try to combine historical map evidence with OSL dating of scroll bars to extend migration rates back in time over the historical period, rather than just much longer periods (as more common with most fluvial OSL applications). The overall approach is sound but some details and assumptions need to be clarified or considered in more detail. My comments concern the historical maps and the meander development, not the OSL. It would be useful to have a summary Figure at the end plotting migration rates themselves over time (with error bars) since much of the discussion is about these, though it can be inferred from Fig. 7. Specific comments: 40 not entirely true that historical maps only cover "cultivated" areas. 53 and elsewhere on p 2 - other literature should be acknowledged, e.g. Rowland et al. (2005). Need to be very careful in naming only one exception that literature search is absolutely comprehensive. 87 give width of channel so we understand its size - very important for subsequent evidence and analyisis. Fig.1 and 120 - cannot refer to a paper that is only submitted and not available. Section 3.2 Need a fuller review of use of historical maps. Hooke and Kain (1982) wrote a book on use of historical evidence, including guidelines on checks to be made. Other subsequent papers give more detail on accuracy in relation to meander changes. Not all older maps necessarily less accurate. Certainly those pre 1840 in Britain were less accurate and not usable for river planform position but late 19thC / early 20thC OS maps at large scale were of high accuracy. 3.2.1 Discuss scale of maps. 140 - "normalization" is a term to be avoided since straightening is far from normal and is completely artificial ( although it is common translation). Use channelization. 150 Finding GCPs is a frequent problem in analysing rivers because of lack of fixed features in floodplains. Is not simply due to nature of maps nut nature of floodplain landscapes. 3.3.1 Explain position of samples in relation to scroll bars and ridges and swales. Precision of 3m seems unnecessarily low, Discuss effects on errors. 3.4 Indicate the width of scroll bars. It is possible to plot the trajectory of maximum meander movement if the scroll bars are highly visible on air photo or satellite images. This would be an added check and corroboration. Also indicates direction of meander movement in relation to the assumed profiles taken. 299 Explain in more detail why or what is meant by assumption of randomness in deposition. 312 but position only accurate to $\pm 3$m 318 Channel width is very likely to have varied significantly over the period of the last few centuries. Numerous papers document such changes on European rivers. Thus assumptions about width introduce another uncertainty, which is assessed but the variations could be real. Again, could supplement with measurements of meander scroll widths if visible on images. 323 Danger of circular argument. Need to be very careful about assumptions

on migration rates since it has been shown that rates tend to accelerate with curvature and during bend development (Hickin, Hickin and Nanson, Hooke, numerous papers). 4.1 I found that maps earlier than 1840 tend to be much less accurate, including the 1st Ordnance Survey maps In England. May help to show presence and absence of features but not exact position. Agree with strategy to exclude the older maps (L439). 417 & 464 state that river moves outside valley but river must be in the valley . Do you mean outside floodplain and moving into terraces and/ or valley wall? If extends into such materials they tend to be more resistant than alluvium so tend to impede channel movement, whereas development shown on Fig. 5 is relatively rapid. Need to explain what the channels move into and what was restricting them prior to that.

---

## Editor Comment (EC1) · J. Braun (Editor) · 14 Jun 2018

I concur with the two reviewers that the content of this manuscript is of great interest and presents a method that is very notative and should provide important constraints on the dynamics of meandering channels. The work is also very well presented and only requires minor to moderate improvements before being accepted for publication in ESURF. Both reviewers provide a list of points that need to be addressed and I strongly recommend that the authors prepare a revised version taking these into account. They include simple suggestions on the style and completeness of references, but also im-

portant points concerning the presentation of the methodology and the results, details about the data used (maps, etc.) and the uncertainty/error associated with them.

---

## Author Comment (AC1) · 5 Jul 2018

Dear Editor and Reviewers,

Thank you very much for your kind words and appreciation of our manuscript. We will take your constructive comments into account, which form a valuable addition. Thank you for your detailed suggestions regarding text and style, we agree and all related points will be included in the revised version. Please find our continued reply below.

[Figure]
**1.** Line 42 I suggest you add "and river management" after "land use". I am thinking of things such as weirs, mills, cut-off channels, bank modification etc.
*We agree; this is what we meant with direct interference with the river; we have clarified the text to make this clear.*

**2.** Line 53 There are some other papers that do this, e.g. including Kemp and Rhodes (2010) QSR, though I am not suggesting you include the citation.
*We will include the citation, as well as a paper by Rowland et al., 2005.*

**3.** Line 70 I very firmly agree that this is underexplored.
*Thank you.*

**4.** Line 138 This is an excellent selection of maps.
*Thank you.*

**5.** Line 140 Is "normalization" a recognised term here? I would possibly suggest "regularization" (not great) or perhaps simply "channel management works" (better).
*We have changed it to "channelization", in line with the suggestion by Janet Hooke (see comment 24).*

**6.** Line 145 It is not clear to me what "otherwise used" includes.
*We have changed the sentence to improve clarity.*

**7.** Line 175 It would be informative to know the values of what the range of spatial errors was here.
*The spatial errors (RMSEs) are available in Table 1, we have added a reference to this*

*table at the end of the sentence.*

**8.** Line 203 "The moment: : :" is not 100% accurate, as some electrons are emitted and recombine shortly after the light source is turned on. I would simply say "When" which has a lesser degree of suggestion of an instantaneous effect.
*We agree and have changed the sentence.*

**9.** Line 204 As partial and incomplete bleaching are a feature of some samples, I would try and be more precise here. I suggest you say something like "the OSL signal is reduced (bleached) to a low level, often close to zero".
*We agree and have included your suggestion.*

**10.** Line 244 It is useful to include the concentration and time of treatment for the HF here.
*We have included details on the HF treatment.*

**11.** Line 260. We need a reference to the measurement conditions, which are presented in Table 2. It is useful to describe the equipment used and the filters used (probably U340 on a Riso set, but let's be sure). Note, in Table 2, "10s cutheat to 180 deg. C" doesn't make full sense. A cutheat by definition is for held for no time. I appreciate that the heat treatment in the second part of the SAR cycle is often referred to as a cutheat to distinguish it from the preheat before the main OSL measurement, so perhaps you could either just use "10s preheat at 180" here, or if you want to preserve cutheat, say "10s "cutheat" at 180"?
*We agree; indeed we used a heating of 10s for the testdose, but like to preserve "cutheat" to make a distinction with the preheat prior to measurement of the natural and regenerative signal. We therefore added the quotation marks as you suggested.*

**12.** Line 322 The use of +/- to mean "approximately" is a little misleading, and not recognised scientific usage. Either use the "$\sim$" symbol, say "approximately". I was misled when I read this, as I assumed this was an uncertainty value on the width, rather than the width value used. In fact, as you used this width value, there is not really

any uncertainty on the value, only on whether it is appropriate (which I think it is!), so I would just say 30m here.
*We agree and have changed all occasions where +/- was used to indicate "approximately" to the symbol ∼.*

**13.** Line 335 It would be useful to know how much uncertainty in position there was here. Give us some typical or max/min values possibly.
*We have added the maximum RMSE value and referred to Table 1, which lists the RMSE values for all maps used.*

**14.** Lines 399-401. Some of this information is repeated from a few line above, so you can contract this first sentence a little.
*We agree and have reduced the text here.*

**15.** Lines 440-450 Is it possible that the channel widths varied significantly, for example during the progression downstream of a gavel slug?
*There are indications that channel width varied through time (Candel et al., 2018), however this will have only have a minor effect on our analysis as the model has low sensitivity to reasonable changes in the z-values (as demonstrated by our sensitivity analysis). We included an additional paragraph in the discussion to address effects of spatial uncertainties.*

**16.** Table 3 and elsewhere. Did you consider using CE (Common Era) rather than AD?
*We agree that using CE is better and thank the reviewer for this valuable suggestion. We have changed all writing of BC/AD to BCE/CE notation.*
**17.** It would be useful to have a summary Figure at the end plotting migration rates themselves over time (with error bars) since much of the discussion is about these, though it can be inferred from Fig. 7.
*Indeed it would be interesting to present such a plot, and we considered it at an earlier stage, and again following the reviewer's suggestion. Nevertheless, we decided not to include such a graph as it can already be deduced from the slope of the graph in Fig. 7. We feel that plotting this information separately (including uncertainty) would be of limited added value, as the precision and resolution of our chronological data does not allow inferences on fluctuations in migration rate. We deduce the average migration rate from Fig. 7 but recommend caution in interpreting rate fluctuations.*

**18.** 40 not entirely true that historical maps only cover "cultivated" areas.
*We agree and have modified the sentence.*

**19.** 53 and elsewhere on p 2 - other literature should be acknowledged, e.g. Rowland et al. (2005). Need to be very careful in naming only one exception that literature search is absolutely comprehensive.
*We have changed the sentence and included the citation.*

**20.** 87 give width of channel so we understand its size - very important for subsequent evidence and analysis.
*We added channel width.*

**21.** Fig.1 and 120 - cannot refer to a paper that is only submitted and not available.
*This paper is currently available in interactive discussion on ESurfD (https://www.earth-surf-dynam-discuss.net/esurf-2018-31/), we have changed the reference to the paper.*

**22.** Section 3.2 Need a fuller review of use of historical maps. Hooke and Kain (1982) wrote a book on use of historical evidence, including guidelines on checks to be made. Other subsequent papers give more detail on accuracy in relation to meander changes.

[Figure]

Not all older maps necessarily less accurate. Certainly those pre 1840 in Britain were less accurate and not usable for river planform position but late 19thC / early 20thC OS maps at large scale were of high accuracy.

*We have added references to general background information on using historical sources (including the book by Hooke and Kain, 1982). We added more references on the maps we used, as the historical background on their production and information on cartographers, survey techniques and depiction methods were addressed in previous publications. For checking map quality we rely on these publications and on our analysis with the MapAnalyst software, which is one of the most recent (2011) techniques for checking map accuracy and can be used in combination with modern GIS techniques.*

**23.** 3.2.1 Discuss scale of maps.
*We have changed the sentence that mentions differences between the maps to include scale.*

**24.** 140 - "normalization" is a term to be avoided since straightening is far from normal and is completely artificial ( although it is common translation). Use channelization.
*We have changed the wording to "channelization".*

**25.** 150 Finding GCPs is a frequent problem in analysing rivers because of lack of fixed features in floodplains. Is not simply due to nature of maps nut nature of floodplain landscapes.
*We agree and have changed the text for clarification.*

**26.** 3.3.1 Explain position of samples in relation to scroll bars and ridges and swales.
*We have added two sentences to the manuscript text to explain the choice of sampling location and distribution of samples over the scroll bar deposits.*

**27.** Precision of 3m seems unnecessarily low, Discuss effects on errors.
*The GPS equipment we used had a precision of up to 3 m. We averaged the precision as recorded with our coordinates, which is 5 m. As sample spacing was 200 m, the*

*effects of these uncertainties on final rate estimates and trends are negligible compared to other uncertainties. As demonstrated by our sensitivity analysis, the model has low sensitivity for minor changes in the z-value (GPS position, channel width). We have added a paragraph in the discussion to address effects of spatial uncertainties.*

**28.** 3.4 Indicate the width of scroll bars. It is possible to plot the trajectory of maximum meander movement if the scroll bars are highly visible on air photo or satellite images. This would be an added check and corroboration. Also indicates direction of meander movement in relation to the assumed profiles taken.
*The width of the scroll bars is indicated in section 3.4 and used to calculate the k0-value for the model (77 and 42 m, for Junner Koeland and Prathoek respectively). The profile of Candel et al. (2018) was taken along a line perpendicular to the point bars. We have added this information to the section on the lithological survey, and refer to the Candel et al. (2018) paper for further details.*

**29.** 299 Explain in more detail why or what is meant by assumption of randomness in deposition.
*Randomness relates to the Poisson process; we mean that lateral migration rates can vary. We have added "varying in rate" between the brackets that were already in the sentence.*

**30.** 312 but position only accurate to 3m
*We have changed the wording and added the precision of the GPS (see also reply to comment 27).*

**31.** 318 Channel width is very likely to have varied significantly over the period of the last few centuries. Numerous papers document such changes on European rivers. Thus assumptions about width introduce another uncertainty, which is assessed but the variations could be real. Again, could supplement with measurements of meander scroll widths if visible on images.
*See reply to comment 15.*

**32.** 323 Danger of circular argument. Need to be very careful about assumptions on migration rates since it has been shown that rates tend to accelerate with curvature and during bend development (Hickin, Hickin and Nanson, Hooke, numerous papers). *We are well aware of this challenge and prevented circular reasoning through the iterative approach, which we believe provides an eloquent solution to circumvent the problem.*

**33.** 4.1 I found that maps earlier than 1840 tend to be much less accurate, including the 1st Ordnance Survey maps In England. May help to show presence and absence of features but not exact position. Agree with strategy to exclude the older maps (L439). *Thank you.*

**34.** 417 & 464 state that river moves outside valley but river must be in the valley . Do you mean outside floodplain and moving into terraces and/ or valley wall? If extends into such materials they tend to be more resistant than alluvium so tend to impede channel movement, whereas development shown on Fig. 5 is relatively rapid. Need to explain what the channels move into and what was restricting them prior to that.
*The investigated meanders have moved outside the former valley and through the former valley side, thereby expanding the floodplain and moving into Pleistocene fluvial deposits and cover sands, locally with Holocene drift-sands on top (all unconsolidated sediments). For clarification we added some lines on this in the section Study Area and emphasised the nature of the sediments reworked by the laterally migrating channel in line 417. The reason of prior restriction within the former valley side is partly addressed by Candel et al. (2018) and is the topic of our next manuscript, so please stay tuned!*